# Phylogenomic analyses highlight innovation and introgression in the continental radiations of Fagaceae across the Northern Hemisphere

Biao-Feng Zhou [1,2,5], Shuai Yuan [1,5], Andrew A. Crowl [3,5], Yi-Ye Liang [1], Yong Shi [1], Xue-Yan Chen [1], Qing-Qing An [1], Ming Kang [1,4], Paul S. Manos [3✉] & Baosheng Wang [1,4✉]

Northern Hemisphere forests changed drastically in the early Eocene with the diversification of the oak family (Fagaceae). Cooling climates over the next 20 million years fostered the spread of temperate biomes that became increasingly dominated by oaks and their chestnut relatives. Here we use phylogenomic analyses of nuclear and plastid genomes to investigate the timing and pattern of major macroevolutionary events and ancient genome-wide signatures of hybridization across Fagaceae. Innovation related to seed dispersal is implicated in triggering waves of continental radiations beginning with the rapid diversification of major lineages and resulting in unparalleled transformation of forest dynamics within 15 million years following the K-Pg extinction. We detect introgression at multiple time scales, including ancient events predating the origination of genus-level diversity. As oak lineages moved into newly available temperate habitats in the early Miocene, secondary contact between previously isolated species occurred. This resulted in adaptive introgression, which may have further amplified the diversification of white oaks across Eurasia.

[1] Key Laboratory of Plant Resources Conservation and Sustainable Utilization, South China Botanical Garden, Chinese Academy of Sciences, 510650 Guangzhou, China. [2] University of the Chinese Academy of Sciences, 100049 Beijing, China. [3] Department of Biology, Duke University, Durham, NC 27708, USA. [4] Center of Conservation Biology, Core Botanical Gardens, Chinese Academy of Sciences, 510650 Guangzhou, China. [5]These authors contributed equally: Biao-Feng Zhou, Shuai Yuan, Andrew A. Crowl. ✉email: pmanos@duke.edu; baosheng.wang@scbg.ac.cn

Northern Hemisphere forests and shrublands are now dominated by species comprising temperate and subtropical lineages, marking one of the major floristic transitions in the vegetation history of the Cenozoic[1–3]. Paleobotanical reconstructions suggest that a cooling global climate afforded ecological opportunities to plant groups that were physiologically predisposed to disperse into and radiate within broadening seasonal biomes across what would become the Americas and Eurasia[4–8]. Central to this pattern of floristic replacement with significant ecological consequence are the roughly 900 species currently recognized within Fagaceae (oak, beech, chestnut, stone oak). Important components of the timing and pattern of macroevolutionary events and the role of ancient hybridization, however, have yet to be sufficiently described across Fagaceae.

The oak family plays a major ecological role in terms of sheer abundance of standing biomass[6,9–17] and a variety of mutualistic associations involving ectomycorrhizal fungi[18–20], gall-forming insects[21–23], and seed-dispersing vertebrates[24–29]. Interactions between Fagaceae and their co-distributed biota suggests degrees of host specificity and the potential for co-evolution, reciprocal diversification, and expansion of range size.

Fossils of modern Fagaceae are well represented in the Northern Hemisphere, indicating long-term presence and differential patterns of diversification[30–40]. Recent studies integrating these fossils within phylogenies of modern taxa have provided essential context to estimate divergence times[41–43]. While a minimum divergence age of ca. 80 million years ago (Ma) is estimated for the family, divergence of crown groups appears to have occurred rapidly in the early Cenozoic, suggesting the potential for rapid morphological change in forest tree species[44–47]. However, the diversification history of Fagaceae remains incompletely understood, with the exception of modern lineages of Quercus[42,48,49]. Therefore, a complete historical account of this continental radiation is needed to bring to light the dynamics of speciation through the genomes of these ecologically important tree species.

Oaks have a long history of divergence in spite of gene flow. Recent estimates of phylogeny using next generation sequencing of nuclear DNA resolve the main oak groups while demonstrating that oak species are generally not of hybrid origin[42,50]. However, more targeted phylogenomic studies have shown that ancient hybridization results in unstable lineages while recent-generation hybrids often fall into intermediate phylogenetic positions between parental lineages[51–54]. New insights into nuclear genomic architecture of hybridization complement various datasets derived from the maternally-inherited plastome and suspected cases of plastome capture and resulting cytoplasmic-nuclear discordance have been shown at various phylogenetic depths in Quercus[55–57]. Now the timing and impact of these events within Quercus, as well as within and between other lineages, is within reach: chronograms for both genomes along with thorough interrogation of the nuclear genome provides the framework needed to estimate the timing of hybridization events, identify the signatures of gene flow, and detect evidence for adaptive evolution.

Phylogenomic analyses of nuclear and plastid genomes reveal a complex history of divergence and gene flow in deep time across Fagaceae. To test specific hypotheses of ancient hybridization, we constructed time-calibrated phylogenies to pinpoint major divergent and reticulate events across a broad sample of 122 individual plants representing 91 species from all recognized genera, using 2124 nuclear loci and full plastomes (Supplementary Data 1 and 2). With these data, we characterize the diversification of Fagaceae and identify admixed genomes due to ancient gene flow within a broad phylogenetic context.

## Results and discussion

**Time-calibrated phylogeny based on nuclear data.** Maximum likelihood (ML) and Bayesian analyses of the concatenated dataset and species-tree analyses using ASTRAL-III and SVDquartets produced similar trees with strong support (bootstrap support (BS) > 90% and Bayesian inference (BI) > 0.95) for all but a few branches (Fig. 1 and Supplementary Fig. 1). All genera of Fagaceae were inferred to be monophyletic with fully resolved interrelationships. Our phylogenetic estimate unambiguously supports three successively sister lineages of Fagaceae—Fagus, Trigonobalanus, and two castaneoid lineages, Castanea + Castanopsis—along with a novel resolution for a crown clade comprising the three remaining castaneoid genera, Chrysolepis, Lithocarpus, and Notholithocarpus, which in turn is sister to Quercus (Fig. 1 and Supplementary Fig. 1). Resolution of castaneoid taxa (Chrysolepis, Lithocarpus, and Notholithocarpus) as sister to Quercus settles long-standing questions on the origin of the wind-pollinated oaks: they are derived from insect-pollinated ancestors that already possessed a single rounded fruit seated within a valveless cupule[58]. Within Quercus, our analyses confirmed the phylogenetic structure resolved by previous studies based on RAD-seq data[42,54] and nuclear loci[51]. Despite phylogenetic congruence across methods, high levels of gene-tree conflict within the nuclear genome were observed, likely due to incomplete lineage sorting (ILS; Supplementary Figs. 2, 3, and 4). This would be expected given the rapid evolution of crown clade genera as inferred here (see below).

We constrained nodes with eight fossil calibrations (Fig. 1 and Supplementary Table 1) to estimate divergence times and diversification dynamics within Fagaceae. Fagus and Trigonobalanus originated by the late Cretaceous, diverging at 81.6 Ma (95% confidence interval (CI) = 82.0–81.1 Ma) and 69.8 Ma (95% CI = 72.6–66.9 Ma), respectively (Fig. 1). Subsequent branching events in the early Cenozoic suggest that the six genera (Castanea, Castanopsis, Chrysolepis, Lithocarpus, Notholithocarpus and Quercus; the hypogeous seed or "HS" clade hereafter) that comprise 98.8% ($N = 893$) of the modern species originated during the Paleocene. The ancestor of the HS clade split at 64.5 Ma (95% CI = 67.0–62.1 Ma) followed by the rapid origination of extant genera within a 15 Ma window (Fig. 1). These events follow the Cretaceous-Paleogene (K-Pg) boundary dated at 66 Ma[59].

Accelerated diversification following the K-Pg mass extinction event has been documented in plants[60,61], birds[62], frogs[63], fish[64], and mammals[65], most likely a generalized consequence of ecological opportunities following the mass extinction. An increase in speciation rate just after the K-Pg boundary was confirmed for Fagaceae by diversification rate analyses, with a net speciation rate shift detected along the branch, leading to the ancestor of the HS clade (Fig. 1). This result is robust to different calibration sets and reference trees for molecular dating (Supplementary Fig. 5).

**Ecological correlates of diversification.** The HS clade shares the derived feature of hypogeous germination, as defined by the first leaves of the embryo remaining in the seed as storage organs that contribute to enhancing seedling survivorship[27]. This condition is often correlated with larger seeds that are biotically dispersed by various specialized animal groups whereas the two successively sister lineages, Fagus and Trigonobalanus, share the plesiomorphic condition of smaller seeds and the generalized state of epigeal germination[66]. Previous phylogenetic studies including fossils have revealed several transitions to biotic dispersal across fagalean lineages during its ca. 95 million-year history[67,68]. Biotically dispersed lineages have larger range sizes and higher

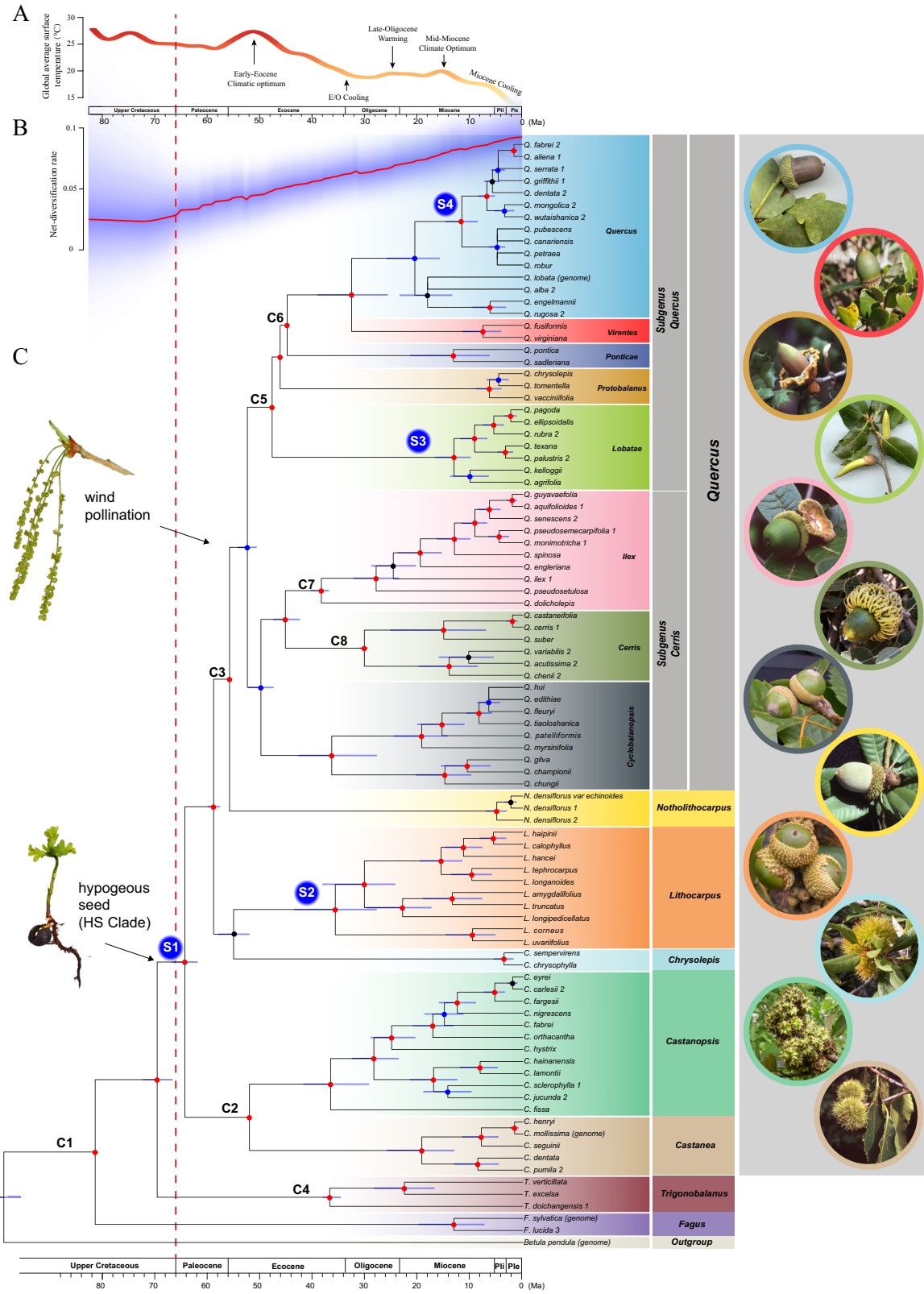

diversification rates than abiotically dispersed lineages. Innovations associated with seed morphology coincide with an increase in diversification rate of the HS clade after the K-Pg boundary (Fig. 1). Time-calibrated phylogenies of the main groups of modern HS seed dispersers, specifically scatter-hoarding Sciuridae (squirrels), Covidae (jays), and Picidae (woodpeckers)

contrast sharply. Evolution of rodent-mediated dispersal closely follows the origin of the HS clade and other large-seeded biotically dispersed fagalean lineages supporting a generalized co-evolution with Sciuridae[65,69–72]. In contrast, the relative timing for the diversification of bird lineages associated with the dispersal of HS seed is at least 20 million years later[73,74].

**Fig. 1 Phylogenetic relationships and divergence time estimation of Fagaceae inferred from analyses of 2124 nuclear genes. A** The global climate curve during the last 82 million years (modified from ref. [8]). Major climate events were indicated. **B** Rate-through-time plot showing the net diversification rate (species/million years) of Fagaceae. Red line is the median and the blue shadow represents the 95% confidence interval. **C** Chronogram derived from ASTRAL-III tree based on concatenated nuclear data. Nodes showing consistent relationships between ASTRAL-III, SVDquartets, maximum likelihood, and MrBayes are marked with red (phylogenetic support ≥ 95% in all four analyses) and blue (support < 95% in any one of the four analyses). Nodes showing conflicting relationships among analyses are marked with black dots. Light blue bars on nodes represent 95% confidence intervals of divergence time estimates and dashed vertical red line represents the age of the Cretaceous-Paleogene boundary (66 million years ago). Geological timescale is shown at bottom. Fossil calibration nodes are indicated with C1–C8 (stem calibration node; Supplementary Table S1). S1–S4 indicate four nodes where shifts in diversification rate were identified. Taxonomic labels of genera, subgenera and sections follow refs. [48, 58, 120]. Illustrations: lax catkins indicate the placement of the change from insect-pollination to wind-pollination that diagnoses the genus *Quercus*; hypogeous seed and seedling marks the origin of the HS clade. Images: representative cupule types are shown on the right. A consistent color scheme was used for taxonomic labels and image borders. Ma, million years ago; Pli, Pliocene; Ple, Pleistocene.

This suggests a second phase of mutualistic response driven by HS seed production generated patterns of co-distribution between granivorous birds, best exemplified in *Quercus*, that likely began during the Miocene[25,75].

Ecological success of Fagaceae is often attributed to symbiosis with at least three main ectomycorrhizal (ECM) lineages of basidiomycetes: Russulales, Boletales, and Agaricales[76]. This mutualism represents an ancient resource-sharing mechanism that contributes heavily to ecosystem processes and dominance of Fagaceae[18,77,78]. The estimated number of global ECM fungal species is c. 6000 with Fagaceae accounting for 45% of the associated 2000 species of host seed plant diversity[76]. While stem lineages of the main ECM clades date back to the Jurassic, crown clade diversification and inferred shifts in speciation rate occur contemporaneously in many of the lineages associated with Fagaceae[79]. Multiple increases in speciation rate postdate the K-Pg boundary by at least 20 million years, suggesting that transition to Fagaceae forests in the Oligocene may have contributed to species radiations of Fagaceae linked to symbiosis. Indeed, secondary increases in speciation rate spanning the Oligocene and Miocene were detected in three clades, *Lithocarpus* from southeast Asia, the Eurasian subclade of section *Quercus*, and section *Lobatae*, which is endemic to the Americas (Fig. 1 and Supplementary Fig. 5). Previous studies based on global sampling of oak species reported four shifts of diversification during the Miocene[42], including the two events we observed within *Quercus*.

Rapid radiation of the genus *Quercus* is coincident with global temperature cooling associated with the onset of temperate habitats during the Oligocene (Fig. 1A). Our phylogenetic analyses confirm that *Quercus* evolved from within a clade formed by all five insect-pollinated castaneoid genera, and diverged from them approximately 56 Ma (Fig. 1 and Supplementary Fig. 5). Fossilized pollen assignable to modern oak sections is found at high latitudes well before *Quercus* migrated to middle latitudes[33]. Thus, the origin of wind-pollination in *Quercus* preceded the explosive radiations of oaks in the Oligocene to early Miocene. We hypothesize the shift to wind-pollination alone did not increase the diversification rate of oak species immediately, but instead served as a predisposed neutral change that later facilitated rapid radiation of this genus during the expansion of seasonal climates (Fig. 1). This hypothesis is partially supported by the findings that oaks have their highest species richness in cool-temperate areas in middle latitudes and montane areas at lower latitudes of the Americas, where they form ecologically dominant forests[6,50].

**Ancient hybridization explains cytoplasmic-nuclear gene-tree conflict.** Plastome-based analyses using various phylogenetic methods (ML or Bayesian analyses), data partitioning schemes (un-partitioned or partitioned by gene and codon position), and alignments (nucleotide or amino acid sequences) yielded largely congruent topologies (Supplementary Fig. 6). Major nodes along the backbone of the plastid tree were highly supported (BS > 80% and BI > 95%) and consistent with the nuclear trees in the placement of *Fagus* and *Trigonobalanus* lineages (Fig. 2). The plastome topology, however, differs markedly from the trees obtained with nuclear loci in regards to the composition and placement of major lineages within the HS clade (Fig. 2). While most plastome subclades comprise related species, several combine disparate taxonomic groups. We failed to recover monophyly of two genera, *Quercus* and *Notholithocarpus*, and six sections of *Quercus* (*Quercus*, *Virentes*, *Ponticae*, *Protobalanus*, *Ilex* and *Cerris*). The structure of the plastome reconstruction within the HS clade is largely geographic, consistent with previous studies[56,57,80,81], with the taxonomic diversity divided into two major clades we treat here as New World (NW) and Old World (OW) (Fig. 2).

The NW-OW pattern recovered in our plastome analyses suggests an early geographic homogenization of cytoplasm across lineages generating the observed cytoplasmic-nuclear discordance at the deepest level of the HS clade (Fig. 2). While the most likely source of cytoplasmic-nuclear discordance is hybridization, incomplete lineage sorting (ILS) could produce a similar pattern. To discriminate between these two hypotheses, we performed coalescent-based simulations. We found the plastid tree discordance to be significantly higher than the expected distribution under a strict coalescent process (Supplementary Fig. 7) and conflicting plastid bipartition frequencies at or near zero in the 10,000 simulated organellar gene trees (Supplementary Fig. 8). ILS alone is therefore insufficient to explain the observed cytoplasmic-nuclear incongruence recovered in these datasets and a scenario of historical gene flow must be invoked.

Hybridization is a widespread phenomenon within modern lineages of Fagaceae, especially between species within sections of *Quercus*[82], and plastome capture events are well documented between sympatric species[56,83–85]. Hybridization is also prevalent between closely related species across many genera within other fagalean families[86–88]. However, the inference of gene flow between modern genera has been based solely on plastome data. When we applied molecular dating methods to the full plastome data, we found estimated divergence times for the deepest splits to generally fall within the rapid diversification phase for the HS clade based on nuclear data (Fig. 1 and Supplementary Fig. 9). Without invoking non-sexual processes such as transmission between incompatible species through intimate physical contact, e.g., plant-plant parasitism and natural root grafts[89], ancient hybridization is the most likely source of deep cytoplasmic-nuclear conflict in Fagaceae. Taken together, our results indicate this pattern of geographic division of reciprocally monophyletic plastome types is best explained as a vestige of widespread ancient hybridization among ancestral populations of the HS clade that became spatially isolated by paleogeographic barriers to gene flow dated minimally to the early Paleocene (Fig. 2 and Supplementary Fig. 9).

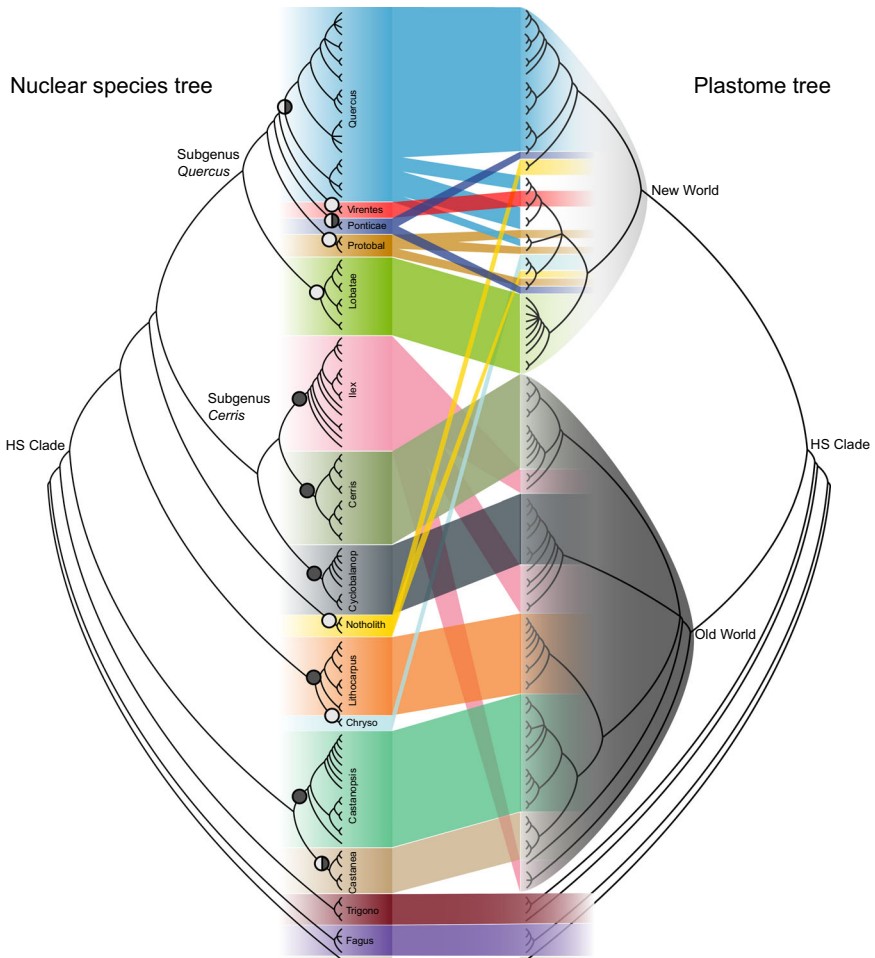

**Fig. 2 Conflicts between nuclear (left) and plastome (right) species trees.** Pie charts on nodes indicate the geographic distribution of the clade (black = Old World, white = New World). The hypogeous seed (HS) clade consists of six genera divided into two major plastome clades: New World (light gray) and Old World (dark gray). Lineage colors are consistent with the color scheme in Fig. 1. Protobal, section *Protobalanus*; Cyclobalanop, section *Cyclobalanopsis*; Notholith, *Notholithocarpus*; Chryso, *Chrysolepis*; Trigono, *Trigonobalanus*.

We additionally found evidence for more recent plastome capture events resulting from hybridization within *Quercus* between the late Miocene to Pliocene (Fig. 2 and Supplementary Fig. 9). As expected, the pattern of discordance between plastome and nuclear genomes uncovers multiple instances where species from phylogenetically distinct clades, but overlapping geographic ranges, share plastome types, for example between species of sections: *Ilex* and *Cerris*, *Virentes* and *Quercus*, *Protobalanus* and *Quercus*, and *Ponticae* and *Quercus* (Fig. 2). While inferring ancient hybridization events using cytoplasmic-nuclear gene-tree conflict provides some evidence of reticulate evolutionary history, satisfactorily confirming and characterizing ancient gene flow requires a detailed investigation into the nuclear genome.

**Ancient gene flow and adaptive introgression detected in the nuclear genome.** Extensive investigation using a *D*-statistic (ABBA-BABA) test detected significant gene flow on 236 (0.911%) of 25882 trios extracted from the species tree ($P < 0.01$ after Bonferroni correction) (Supplementary Data 3). Not surprisingly, most cases of gene flow appeared to be recent in origin and between closely related species from within genera or sections of *Quercus* (Fig. 3A). Ancient gene flow, however, was detected between Eurasian white oaks (section *Quercus*) and *Q. pontica* (section *Ponticae*) and between North American white oaks (section *Quercus*) and the ancestor of section *Virentes* (Fig. 3), consistent with the results of gene-tree

analyses from the two genomes (Fig. 2). Network analyses using SNaQ confirmed historical gene flow between *Q. pontica* and Eurasian white oaks inferred in the current study (Supplementary Fig. 10) and previous studies[51].

We also assessed the distribution of alternative topologies within our 2124 nuclear gene dataset and found introgressed signals to be widely scattered across the genome (Supplementary Fig. 11). This is expected given that long-term recombination tends to fragment introgressed stretches of DNA following initial hybridization events[90]. However, positive selection has been shown to maintain long introgressed haplotypes in populations of humans and maize[91,92], with the length of introgressed fragments increasing with stronger selection[93]. Our investigation of putatively ancient hybridization events between sections of *Quercus* yielded haplotypes that were significantly longer than expected under neutrality. Identity-by-descent (IBD) analyses based on whole-genome SNP data clearly detected a large number of shared haplotype blocks (see Methods) for three lineage-pairs, i.e., *Q. pontica* vs. European and Asian white oaks, North American white oaks vs. section *Virentes*, and North American white oaks vs. *Q. sadleriana* (Fig. 4B–D). However, we did not find IBD blocks of similarly long lengths between other *Quercus* sections in which we documented plastome capture events (Supplementary Table 2).

Within the long sets of shared IBD regions, the *D*-statistic test revealed gene flow between oak sections (Supplementary Fig. 12).

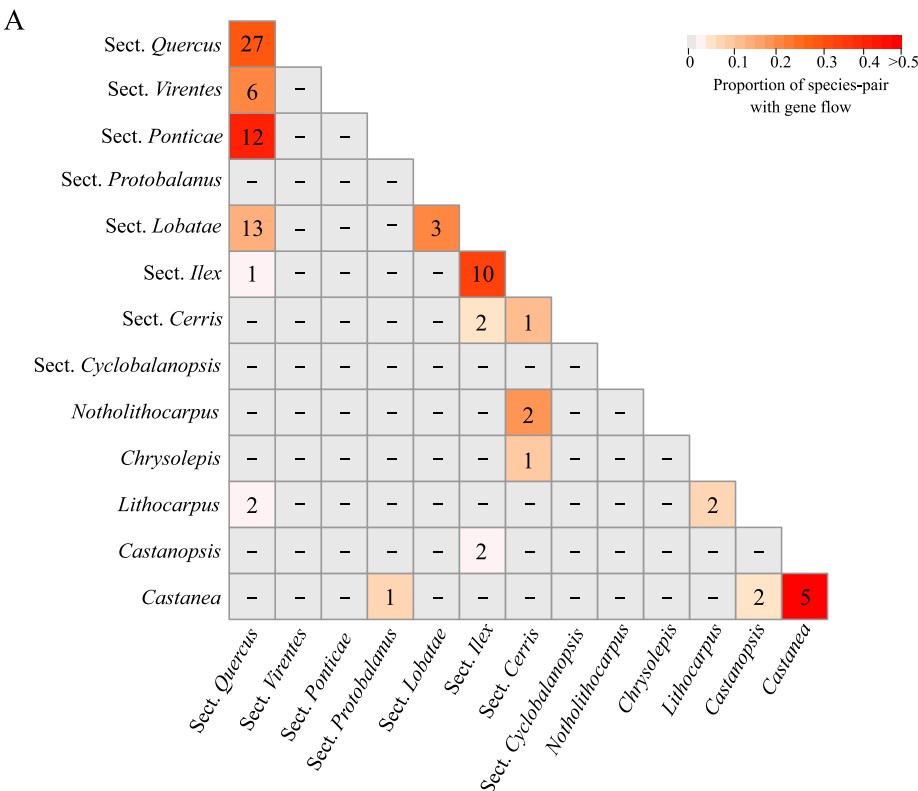

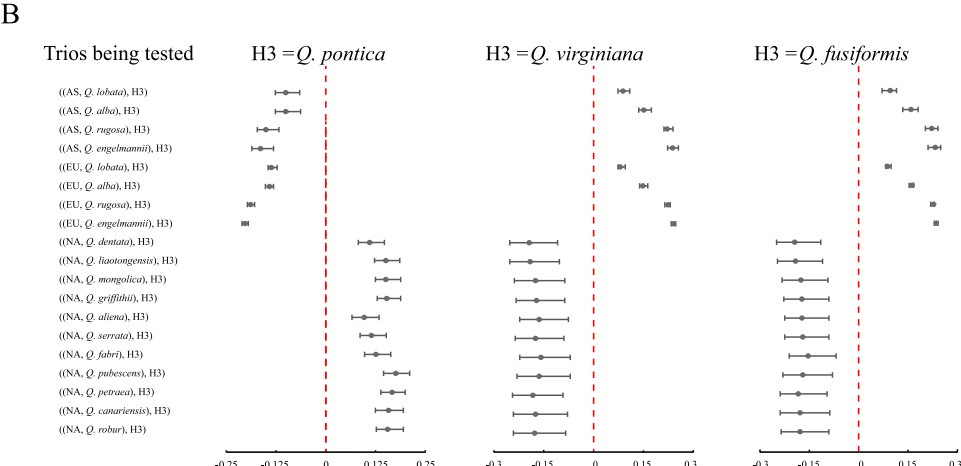

**Fig. 3 Gene flow between Fagaceae species revealed by *D*-statistic test. A** Number of species-pairs with significant *D*-value between sections of *Quercus* and other genera. Numbers on diagonal line indicate gene flow within each section or genus. Cells are colored based on the ratio of species-pairs with gene flow, with warmer colors indicating a higher proportion of species-pairs showing gene flow. For example, significant gene flow was detected for 12 species-pairs between sections *Quercus* (white oak) and *Ponticae*, representing 41% of tested species-pairs between these two sections. **B** Distribution of *D*-values for white oaks vs. *Q. pontica* (left), and two species of section *Virentes*, *Q. virginiana* (middle) and *Q. fusformis* (right). Each line summarizes a set of *D*-statistic tests performed on trios in the format ((H1,H2),H3) with different H1 species and fixed H2 and H3 species (one of the three species above; sample size *n* = 4, 4, and 7 species for H1 as NA, EU, and AS, respectively). Both H1 and H2 were white oaks, but represent different lineages. For example, if H2 was a North American white oak, then H1 was sampled from European or Asian white oaks. In each panel, points represent mean *D*-values and error bars represent minimum and maximum *D*-values across multiple tests. EU = European white oak; AS = Asian white oak; NA = North American white oak. A negative *D*-value indicates gene flow between H1 and H3 while a positive *D*-value indicates gene flow between the H2 taxon and H3. *Q. pontica* shows a clear pattern of gene flow with EU and AS white oaks but not with NA white oaks while the opposite pattern is recovered for *Q. virginiana* and *Q. fusiformis*. The significance of the *D*-value was tested by a two-sided standard block-jackknife procedure implemented in Dsuite v0.3[158] with default parameters, and adjusted by Bonferroni correction for multiple comparisons.

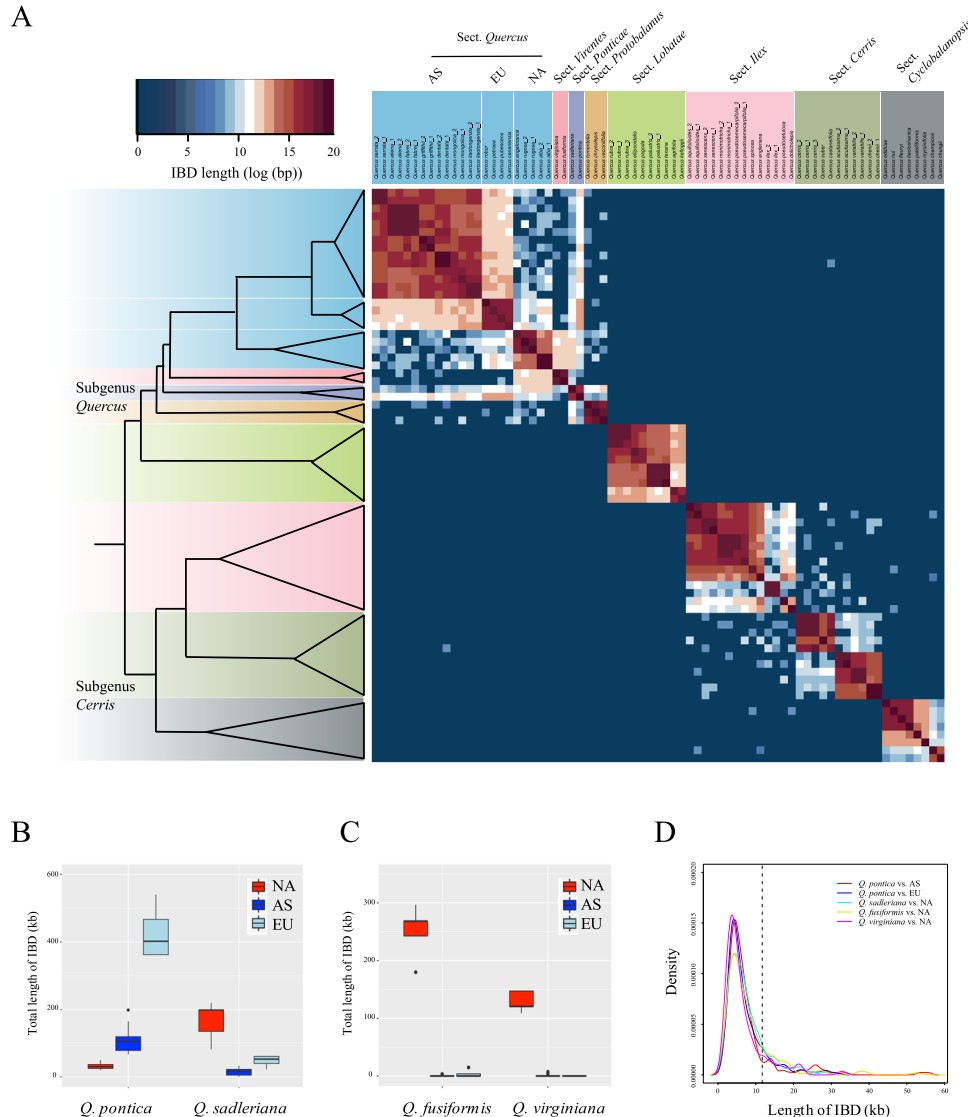

**Fig. 4 Shared IBD blocks between *Quercus* species. A** Heatmap indicating the total length of identity-by-descent (IBD) blocks for each pair of comparisons. **B**, **C** box plots show shared total length of IBDs between sections *Ponticae* and *Quercus*, and between sections *Virentes* and *Quercus*. In these box plots, the horizontal lines indicate the median value, the bottom and top of each box represents the first and third quartiles, and the whiskers extend to 1.5 times the interquartile range (the sample size $n = 4$, 5, and 14 individuals for NA, EU and AS, respectively). NA = North American white oak; EU = European white oak; AS = Asian white oak. **D** Kernel distribution of the length of shared IBD blocks between sections. Vertical black line (at 11,724 bp) indicates the shortest IBD block that is significantly longer than the expectation for selectively neutral introgressed fragments maintained in a population under a constant recombination rate of $10^{-8}$ per site per year, assuming an average divergence time of 3 million years ($P = 0.0476$; two-sided probability estimated under a Gamma distribution function, and adjusted by Bonferroni correction; see details in Methods).

In addition, the recombination rate in the same IBD regions was not different from genomic background ($W$ ranges from 82978 to 321198, $P = 0.66$–$0.73$, Mann-Whitney $U$-test; Supplementary Table 3), and the length of the IBDs was not associated with recombination rate (Spearman's $\rho = -0.19$ to $0.14$, $P = 0.10$–$0.79$; Supplementary Fig. 13). Therefore, these haplotypes shared between *Quercus* sections are most likely due to historical inter-sectional hybridization instead of the maintenance of ancestral polymorphisms in regions with reduced recombination rates. To test this prediction, we calculated the probability of maintaining selectively neutral haplotypes of a given length in both oak sections after introgression using methods developed to study introgression in humans[91] and using generation times and mutation and recombination rates derived for oak species[94–96]. We determined that 166 IBD blocks (11724–113757 bp) were significantly longer than expected if the introgressed fragments

were selectively neutral ($P < 0.05$; Fig. 4D; see details in Methods), suggesting that the IBDs identified here provide convincing evidence of adaptive introgression. Multiple Gene Ontology (GO) categories with important metabolic processes and molecular functions (e.g., terpene metabolic processes, sesquiterpenoid metabolic processes) were overrepresented for genes located in these IBD regions (Supplementary Table 4), further suggesting a diverse set of genes and functional categories may have contributed to adaptive introgression of oak species. Adaptive introgression between closely related species has recently been documented in *Quercus*[97–99]. Our study posits that introgressed elements between divergent oak sections could be preserved for millions of years by natural selection.

With the exception of the few cases involving sections *Quercus*, *Ponticae* and *Virentes*, we found no corroborating evidence of hybridization within the nuclear genome of the remaining

lineages exhibiting cytoplasmic-nuclear gene-tree conflict. The occurrence of plastome capture events in the absence of detectable nuclear introgression is not unexpected, and could be due to the early phases of hybrid zone dynamics[100,101]. For example, extensive backcrossing with one parental species after initial hybridization could sweep out signals of reticulation events in the nuclear genome and recombination over long evolutionary time could have degraded signals of ancient hybridization[102,103]. In oaks, backcrossing is preferentially unidirectional[104,105] and linkage disequilibrium typically declines to background quickly[106,107], blurring the signals of past introgression in the nuclear genome. Alternatively, as mentioned above, plastomes can be captured through non-hybridizing means such as intimate physical contact, e.g. plant-plant parasitism and natural grafts[89], which would leave no signal in the nuclear genome.

**Genomic footprints of a changing temperate forest**. We show that the story of the evolution of modern Fagaceae can be told through the lens of two unlinked genomes, each contributing unique perspectives on the complex combination of divergent and reticulate historical events that unfolded through the Cenozoic. Further, historical migration events in temperate lineages are inferred by discovery of three exceptions to the NW-OW plastome pattern (Fig. 2). Chestnuts (Castanea) currently distributed across the Holarctic arose in the OW and moved to the NW, while the modern Eurasian oak sublineages of sections Quercus and Ponticae are NW in origin, consistent with RAD-seq analysis[54]. These bidirectional land bridge crossings with unambiguous origins document the timing of limited, but key dispersal events leading to the spread of modern Fagaceae forests across the Northern Hemisphere[108]. While an untold number of extinctions will escape our detection, the reciprocal migrations of oak and chestnut species during the late Eocene to Oligocene provide evidence for the origins of ecologically significant components of Northern Hemisphere forests. The ecological implications of these biotic exchanges of keystone lineages await future study.

Hybridization, common throughout Fagales, may be adaptive at various stages of diversification depending on patterns of persistent interfertility and range overlap among lineages[109]. For Fagaceae, an early stage of widespread hybridization among ancestral elements of the HS clade is suggested by an enduring paleophylogeographic signal in the plastomes of modern lineages. Soon after, a rapid burst of cladogenesis at the base of the HS clade, tracked by the nuclear genome, generated the extant lineages as resolved here. As reproductive isolation evolved across most of these lineages, divergent evolution generated sets of exclusive plastome haplotypes within the broader phylogeographic pattern observed here, except for the instances where interfertile oak lineages experienced secondary contact during the Miocene. Within several clades of Quercus, cytoplasmic-nuclear gene-tree incongruities support previous studies indicating an expanded role of hybridization in flowering plant evolution[110–113].

We suggest that oaks and their chestnut relatives have been hybridizing for millions of years. In Fagaceae, this is facilitated by small and evolutionarily stable genomes, high levels of synteny, and a consistent chromosome number across taxa[114–119]. In addition to conserved genomes and maintenance of some level of interfertility, these lineages share other life-history traits with diverse and often tropical tree genera that suggest the syngameon is functionally adaptive. Fagaceae species in particular share evolutionary and ecological characteristics that may promote adaptive introgression including generalized pollination systems, high levels of fecundity, and widespread sympatry[44,115,117].

Consequently, we document three main geographic areas of historical introgression between oak sections as evidenced by plastome capture: western North America, southeastern North

America, and Eurasia. These areas are known to be centers of phylogenetic diversity for the genus[42] with extensive zones of sympatry and evidence for convergent evolution of form in response to climate[11,50]. In two of these areas, specifically Eurasia where the ranges of sections Ponticae and Quercus once overlapped and the American southeast where sections Virentes and Quercus are still known to hybridize, we present evidence from the nuclear genome that ancient hybridization has left a signature of adaptive evolution. While more detailed study is necessary to fully appreciate the impact these introgressed alleles may have had on the modern oak landscape in these regions, ancient hybridization between the relictual Q. pontica sublineage of sect. Ponticae and the widespread Eurasian sublineage of sect. Quercus appears to have contributed to an increased diversification rate in section Quercus during the Miocene (Fig. 1; see also Hipp et al.[42].). This uptick in speciation and the ecological opportunity available to the white oaks marks the rise and spread of a dominant deciduous lineage bearing an introgressed nuclear genome into the forested ecosystems across Eurasia.

## Methods

**Taxon sampling, DNA extraction, and whole-genome sequencing**. We constructed a comprehensive Fagaceae dataset consisting of 122 individuals from 91 species representing all eight currently recognized genera[58,120]. Complete taxon sampling was achieved for three small genera: Chrysolepis (2 species), Notholithocarpus (2 species) and Trigonobalanus (3 species). For the remaining genera, representative samples for all major lineages were included: Fagus (2), Castanea (5), Castanopsis (12) and Lithocarpus (10). For the well-studied genus Quercus, extensive sampling (54 species) was conducted to represent all eight recognized sections[42,48]. Several species were represented by multiple accessions collected from different natural populations or cultivated plants. Betula pendula was selected as an outgroup due to its close relationship to Fagaceae and the availability of an assembled genome[121]. Accession information is provided in Supplementary Data 1.

Total genomic DNA was extracted from silica-dried leaf tissue using BioTeke Genomic DNA Extraction Kit (Beijing, China). High-quality DNA was used to constructed paired-end sequencing libraries with an insert size of 500–600 bp according to the Illumina library preparation protocol. Sequencing (150 bp paired-end) was carried out on the NovaSeq platform at Novogene (Beijing, China) to a coverage of 25–40× for all samples.

**Orthologous gene identification and nuclear alignment matrix assembly**. To obtain orthologous genes (OGs) for phylogenetic analysis, we performed a series of critical search and filtering processes. There are four high-quality genome assemblies (chromosome-level) available in Fagaceae: Fagus sylvatica[122], Castanea mollissima[123], Quercus robur[96] and Quercus lobata[124]. These four assemblies together with the outgroup species B. pendula[121] were used to identify putative OGs in OrthoFinder v2.3.12[125] with an E-value of 1E-5. Orthologous groups containing only one sequence from each examined species were retained to minimize paralogs in subsequent phylogenetic analyses. Single copy genes (SCGs) identified by OrthoFinder may still have duplicates, either as pseudogenes or un-annotated functional genes in the genome. To identify and remove additional multiple-copy genes, we blasted coding sequences (CDS) of SCGs against each of the five genomes using BLAST+ v2.10.1[126]. We filtered alignments using the following thresholds: E-value < 1E-5, alignment length ≥ 80% of the query sequence, and identity ≥ 80%. We kept CDS with only one hit in each of the five species. The retained CDS regions identified as belonging to a single gene were concatenated for subsequent phylogenetic analyses.

To generate the nuclear DNA sequences, we sequenced whole genomes of 117 individuals (Supplementary Data 1) to a coverage of 25 − 40× using the Illumina NovaSeq platform and called genotypes in SCG regions. We trimmed and filtered raw reads using Trimmomatic v0.39[127], mapped high-quality reads to a reference genome using BWA v0.7.17[128], and called genotypes via HaplotypeCaller in GATK v4.2[129]. A simulation study found that the inclusion of nonpolymorphic positions in the alignment and mapping short reads to multiple references could improve the accuracy of phylogenetic inference[130]. Thus, we called all available sites (both variants and invariants). To reduce the effects of reference bias, we used three reference genomes for mapping and SNP calling in related species. The genome of F. sylvatica was used as reference for genus Fagus, the Q. robur genome was used for genus Quercus, and the Castanea mollissima genome was used for the remaining six genera. We only considered sites with mapping quality ≥ 30 and base quality ≥ 30, and further filtered variants using the following criteria: (1) homozygous genotypes with depth <4 or heterozygous genotypes with depth <20 were assigned as missing; (2) sites with mean depth <5 or >100 across all individuals were discarded; (3) sites with proportion of heterozygous genotypes >50% were excluded.

To obtain an aligned matrix of SCGs, we generated a 6-way whole-genome alignment based on the four reference genomes and two additional assemblies (*Q. lobata* and *Q. suber*) following a lastZ/Multiz pipeline[131,132]. We used *Q. robur* as a reference genome for genome alignment, and merged genotypes from mapping to different references or extracted from different assemblies together according to their relative positions on the *Q. robur* genome. The data matrix was then filtered by excluding sites containing ≥ 10% missing data, and SCGs with length <200 bp. Alignments with divergent paralogous genes usually show elevated levels of polymorphism, thus we further excluded SCGs with polymorphism in the top 95th percentile (cutoff = 43.8%). Every OG was presented in all sampled individuals with no missing data. Our final dataset included 2124 SCGs with a total length of 1,689,974 bp for data analyses (Supplementary Data 2).

**Evaluating the impacts of reference genomes on the accuracy of SNP calling and phylogenetic reconstruction**. We applied both empirical and simulation analyses to assess the impacts of the reference genome on the accuracy of SNP calling and phylogenetic reconstruction. The assembly of *Castanea mollissima* was used as the reference genome for SNP calling in *Castanea* and the five genera (*Chrysolepis, Castanopsis, Lithocarpus, Notholithocarpus* and *Trigonobalanus*) without available genome assemblies. To test whether reference bias was introduced by using a divergent reference genome, we re-called SNPs for these five genera by using *Q. robur* as reference, and compared genotypes called from *Q. robur* with those from *C. mollissima*. Despite the slightly higher rate of missing data (9.29–9.72%) observed with using *Q. robur* as reference genome compared to *C. mollissima* (3.96–4.27%), 95.62–95.84% genotypes were identical between these two datasets (Supplementary Table 5). Identical tree topologies also were generated based on the two datasets when using the same phylogenetic method (data not shown), suggesting weak reference bias in our data.

To further monitor the accuracy of genotyping in the query dataset with different divergence levels from the reference genome, we generated mutated sequences (henceforth referred to as "mutated-sequence") by randomly adding 0.25%−20% mutations to the longest chromosome of *Q. robur* (chromosome 2, henceforth referred to as "reference-sequence"). Next, we used WGSM (https://github.com/lh3/wgsim) to simulate 150 bp pair-end reads from each mutated-sequence with 30× coverage (close to our sequencing depth 25–40×). Simulated reads were mapped to the reference-sequence, and SNPs were called and filtered using the same protocol as described above. For each simulated dataset, we compared genotype calls to the mutated-sequence from which the datasets were generated. To mimic the real data, SNPs called from the repetitive regions were excluded from data analyses. The true positive (TP) rate was defined as TP/(TP + FP), where TP is position identical to mutated-sequence, and FP (false positives) are called genotypes different from mutated-sequence. The missing rate (MR) was defined as MISS/SIZE, where MISS is non-genotyped sites and SIZE is total sites (~51.2 Mb) in the reference-sequence after excluding masked repetitive regions. High TP rate (>97.7%) and low MR (<1.5%) were found in datasets with divergence levels from reference-sequence no more than 10% (Supplementary Fig. 14). By extracting sequences of the 2124 SCGs from the 6-way whole-genome alignment, nucleotide divergence was estimated as 7.46 − 7.69% between most divergent genera (i.e., *Fagus* vs. *Quercus* and *Castanea*) (Supplementary Table 6), genotypes called from SCGs by using a divergent reference (e.g., using *C. mollissima* for other genera) would not result in strong reference bias.

**Plastome assembly and alignment**. We assembled 117 plastomes during the course of this study and obtained five additional plastomes from Genbank (Supplementary Data 1). Raw reads from whole-genome sequencing were used for de novo assembly of plastomes in NOVOPlasty v4.2[133]. A ribulose-bisphosphate carboxylase (*rbcL*) gene sequence from *Quercus rubra* was used as the seed sequence for assembly. Assembled plastomes were annotated using the program PGA v1.0[134]. The boundaries of inverted repeats and coding regions of each annotated gene were determined in Geneious v7.1.4[135] by using the *Q. rubra* plastome as a reference. Coding regions of 76 protein-coding genes present in all species were extracted from the assemblies (Supplementary Table 7), aligned using MAFFT v7.221[136], and manually adjusted using Bioedit v7.2 (https://bioedit.software.informer.com). Based on plant plastid genetic code, the codon alignment was translated into amino acid sequences. A preliminary phylogenetic analysis found two *Q. ilex* samples were placed as a sister group to all other Fagaceae species except the genera *Fagus* and *Trigonobalanus*. This is likely an artificial clustering, because previous analyses with extensive sampling (26 individuals) spanning the geographic distribution of *Q. ilex* placed this species within a clade formed by Eurasian oaks and genera *Castanea* and *Castanopsis* based on plastid data[57]. Therefore, we excluded these two *Q. ilex* samples from subsequent plastome analyses. Removing these two samples did not change the topology among other species (data not shown). The plastome alignment is 65,814 bp in length, of which 11,058 characters were polymorphic. A list of the 76 genes is presented in Supplementary Table 7. The alignment of nuclear genes and plastomes can be found in Dryad-archived data (https://doi.org/10.5061/dryad.vq83bk3tc).

**Phylogenetic analyses**. Phylogenetic analyses were conducted using Maximum Likelihood (ML) and Bayesian approaches for concatenated nuclear and plastome data. Partitioned ML analysis was performed using RAxML v8.2.12[130]. The best-scoring ML tree was found from 1000 ML trees, and topological robustness was evaluated by using 1000 non-parametric bootstrap replicates. Bayesian analysis was conducted in MrBayes v3.2.6[137]. Markov chain Monte Carlo (MCMC) runs were performed for 10 million generations, and trees were sampled every 100 generations. The first 25,000 (25%) trees were discarded as burn-in to ensure that the chains were stationary. The remaining trees were used to generate a strict consensus tree and to calculate posterior probabilities for each node.

PartitionFinder2 v1.1[138] was used to determine the optimal partitioning strategy and evolutionary model of each partition under the Akaike Information Criterion (AIC)[139]. For nuclear DNA data, partitioning by gene yielded 35 partitions in the best scheme. For plastome DNA data, full partitioning scheme by both locus and codon position (each of the three codon positions in each gene as one partition) was examined, and the best scheme contained 24 partitions. For plastome amino acid data, each gene was considered as one partition, resulting in 12 partitions in the optimal scheme. In ML analyses, the GTRGAMMA model was used for all DNA sequence partitions, and the evolutionary models chosen by PartitonFinder2 were used for amino acid partitions. For Bayesian analyses, the evolutionary model identified by PartitionFinder2 was used for each DNA and amino acid partition. The models, partitions, and alignments used for phylogenetic analysis can be found in Dryad Data Archive (https://doi.org/10.5061/dryad.vq83bk3tc).

Two species-tree analyses were performed. First, we applied a summary method using ASTRAL-III v5.7.3[140]. Gene trees were estimated from single-gene alignments using RAxML with GTRGAMMA model and 1000 fast bootstrap replicates. Individual gene trees (best trees) and bootstrap replicates were used to estimate a species tree in ASTRAL-III with 1000 coalescent bootstrap replicates. Following Zhang et al.[140], branches with low support were removed to improve the accuracy of tree inference. We tested different thresholds by collapsing branches with support <10, 20, 30, 40, and 50%, and obtained near-identical tree topologies (data not shown). The tree generated by ASTRAL-III with 50% threshold is presented.

SVDquartets v1.0[141], a method based on site pattern frequencies and algebraic statistics implemented in PAUP v4.0a152[142] was additionally used to estimate a species tree. This method was originally designed for SNP data, but also performed well on large multiple-locus datasets[141]. The concatenated nuclear data matrix was used as input for SVDquartets. All possible quartets were evaluated, and clade support was assessed using 500 bootstrap replicates.

**Divergence time and diversification rate estimation**. Divergence time estimation was conducted for both plastome and nuclear datasets using MCMCTree v4.9j in the PAML v4.9j package[143]. MCMCTree estimates divergence times using an approximate likelihood method, and is computationally efficient with large genomic data[144]. The MCMC chains were first run for 3 million generations as burn-in, and then were sampled every 400 generations until a total of 25,000 samples were collected (10 million generations). Tracer v1.7 and LogCombiner v1.10 were used to confirm the convergence across each run and ensure the effective sample size of all parameters were greater than 200. For each of the plastome and nuclear datasets, three independent runs with different seeds were compared for convergence, and similar results were generated.

For nuclear DNA data, we divided the 2124 nuclear genes into three partitions according to substitution rates estimated by Baseml v4.9j in PAML with a strict molecular clock and then applied an uncorrelated rate model (clock = 2 in MCMCTree) to infer divergence times. We used priors of G (1, 6.1677) for the overall substitution rates (rgene_gamma), G (2, 5, 1) for the rate-drift parameter (sigma2_gamma). As concatenated and species-tree analyses revealed different relationships among genera *Quercus + Notholithocarpus*, *Lithocarpus* and *Chrysolepis*, we constrained each alternative topology and constructed the ML reference tree for dating. As data heterogeneity may bias the divergence time estimation[145], we also applied two "gene-shopping" methods to identify genes with the best information for dating. First, we used SortaData[146] to filter 212 (top 90th percentile) most clock-like loci by considering clock-like as the primary criterion, followed by tree-like and tree length. Second, we calculated the Robinson-Foulds (RF) distance between gene trees and the reference tree following Johns et al.[147], and retained 212 loci with the lowest RF distance and highest concordant phylogenetic signals. By using three different topologies as reference trees (see above), we generated six reduced datasets (Supplementary Data 4). The divergence time estimated on the 2124 genes were almost identical to the reduced datasets (Pearson's correlation coefficient r = 0.991-0.995, P < 2e⁻¹⁶; Supplementary Fig. 15). For plastome data, we treated all 76 plastome genes as one partition, and estimated divergence times by using the plastome ML tree as reference under an uncorrelated rate model. We set priors of rgene_gamma and sigma2_gamma parameters as G (1, 41.667) and G (2, 5, 1), respectively.

Based on results of Xiang et al.[68], the root age of Fagaceae was constrained to 95.5–101.2 Ma for both plastome and nuclear data. For nuclear data, we further added six additional widely accepted fossil calibrations (Supplementary Table 1). For the plastome analysis, only two calibrations could be used due to non-monophyletic lineages in the plastome tree (Supplementary Table 1).

For comparing the estimated divergence time between plastome and nuclear datasets, we also dated the nuclear tree using the same two calibrations applied to the plastome tree. For species with multiple samples, we chose one individual for dating the nuclear DNA tree, while retaining all individuals for dating the plastome tree because many species were not monomorphic for their plastome.

To estimate the diversification rate of Fagaceae, we applied Bayesian Analysis of Macroevolutionary Mixture (BAMM v2.5.0)[148]. The time tree estimated by MCMCtree was used as an input tree. To account for incomplete taxon sampling, we calculated sampling fraction of each genus and each section of genus *Quercus* based on the number of species recorded in previous reports[42,120], and then added un-sampled taxa to a random position in each corresponding lineage (Supplementary Table 8). The BAMM analyses were run for 10 million generations, saving every 1000 generations. The first 30% samples were discarded as burn-in, and the remaining samples were summarized and plotted using BAMMtools v2.1.5[148].

**Topological concordance analyses.** To evaluate the conflicts between nuclear gene trees and the species tree, we first calculated the internode certainty all (ICA) to quantify the degree of conflict on each node between a target tree and gene trees[149]. ICA values close to 1 indicate strong concordance for the bipartition defined by a given internode, while ICA values close to 0 indicate strong conflict. Negative ICA values indicate that the defined bipartition conflict with other high frequent bipartitions. The ICA values were estimated in RAxML and the species tree found by ASTRAL-III was used as the target tree. We further summarized the number of conflicting and concordant bipartitions with PHYPARTS[150], using the species tree estimated by ASTRAL-III and the individual gene trees.

**Evaluation of substitutional saturation and codon-usage bias within the plastome dataset.** To investigate whether base substitution saturation biased the accuracy of phylogenetic inference in plastome phylogenetic analyses, we estimated the amount of substitution saturation using methods detailed in Xia et al.[151]. This involved employing critical index of substitution saturation (ISSc) that defines a threshold for significant saturation in the data. From the data of 76 plastome genes, we assessed the level of substitution saturation for codon12 and codon3 using the program DAMBE v7.035[152], and found that there was sufficient phylogenetic information at all codon positions (Supplementary Table 9).

To investigate how synonymous codon usage varies among Fagaceae species, and whether synonymous codon biases have resulted in artificial and random phylogenetic inference, we measured Relative Synonymous Codon Usage (RSCU) values using GCUA[153]. RSCU is defined as the ratio of the observed codon appearance to the number expected given that all synonymous codons appear with uniform frequency. We found similar level of GC content and variation in codon bias across Fagaceae species (Supplementary Fig. 16). These results suggested that Fagaceae plastid genomes are highly conserved, and the plastid-based analyses would be not biased due to substitution saturation or compositional heterogeneity among species.

**Coalescent simulation.** To test whether incomplete lineage sorting (ILS) alone could explain the incongruence between plastome tree and nuclear species tree, we followed Folk et al.[154] to simulate 10,000 plastome trees under the coalescent model using DENDROPY v4.1.0[155]. The ASTRAL-III tree was used as a guide tree for the simulation. To simulate plastome trees, branch lengths were scaled by a factor of four to account for the haploidy and maternal inheritance of the plastome. Clade frequencies of simulated trees were summarized using PHYPARTS[150]. In the scenario of ILS alone, the topology from our empirical plastome tree should be present in simulated trees with high frequency; if gene flow is present, the topology recovered in our empirical tree should be absent or at very low frequency in the simulated trees. Following previous studies[103,156], we also counted the number of extra lineages in observed and simulated trees using the function deep-coal_count in Phylonet v2.4[157]. In the case that gene flow is present, more extra lineages are expected in the observed trees relative to simulated trees.

**Gene flow analyses.** To detect potential gene flow between species, we performed ABBA-BABA statistic tests in Dsuite v0.3[158]. These analyses take advantage of a four taxon statement ((H1, H2)H3)H4). With H4 as the outgroup, H1 and H2 are treated as a pair of sister species and H3 is tested as the species with potential gene flow with H1 or H2. The number of sites with allele patterns of ABBA and BABA are tallied. The *D*-statistic is derived from calculating $D = (nABBA - nBABA)/(nABBA + nBABA)$, where nABBA and nBABA are the total number of sites with patterns of ABBA and BABA, respectively[159,160]. A negative *D*-value indicates gene flow between H1 and H3, a positive *D*-value indicates gene flow between H2 and H3, and $D = 0$ indicates no gene flow[159,160]. As ABBA-BABA test assumes a sister relationship between H1 and H2, we restricted our analyses by sampling H1 and H2 from same genera, or same sections within genus *Quercus*. In addition, because H1 and H2 are sister species, the sites with the pattern of BBAA are expected to be larger than ABBA and BABA patterns. We further filtered trios that violated this assumption, and applied ABBA-BABA test to 25882 trios extracted from the species tree. To account for multiple testing, we corrected *P*-values with Benjamini–Hochberg false discovery rate (FDR)[161]. For a pair of species involved

in multiple tested trios (for example, while H2 and H3 are fixed, there may be different H1 taxa available, thus different *D*-values for H2 and H3 may be generated), the estimated *D*-value with lowest FDR was retained. An individual of *Trigonobalanus doichangensis* was used as an outgroup for all tests. To test how outgroup choice influenced the analysis, we also used an individual of *Notholithocapus densiflorus* in tests within *Quercus* and obtained results similar to those using *T. doichangensis* (data not shown).

To further explore the reticulate evolutionary histories within Fagaceae, we inferred species networks using SNaQ[162] implemented in the package PhyloNetworks v0.12.0[163]. SNaQ is a pseudolikelihood method, which estimates a phylogenetic network while accounting for both ILS and gene flow[162]. We reduced the dataset to a computationally tractable size[164], and generated four sub-datasets each with 15–17 taxa sampled. For each sub-dataset, we sampled species showing inconsistent placement between nuclear and plastome trees. The first one focused on relationships within subgenus *Quercus* and a sample of 16 species (Supplementary Fig. 10). The second one focused on the relationship within subgenus *Cerris* and a sample of 15 species (Supplementary Fig. 10). The third one focused on the relationships among genera *Castanea*, *Castanopsis*, *Lithocarpus*, and *Quercus*, and the forth one other focused on the relationship among genera *Chrysolepis*, *Notholithocarpus*, and *Quercus* (Supplementary Fig. 10). One individual gene-tree generated by RAxML were used as input, and nested analyses were performed allowing for zero (*h* = 0) to four (*h* = 4) hybridization events. Each nested analysis was optimized by 10 independent runs, and the best fitting model was selected based on the log pseudolikelihood score.

To investigate the genomic pattern of introgressed loci, we quantified the distribution of phylogenetic signal for conflicting topologies across nuclear gene trees, and then mapped loci supporting alternative partitions to the *Q. robur* genome[96]. Following Shen et al.[165], we calculated site-wise log-likelihood scores for the primary and alternative topologies in our concatenated matrix using the "-f G" command in RAxML. After that, the difference in site-wise log-likelihood scores (ΔSLS) between topologies were summed across sites in each gene, generating gene-wise log-likelihood scores (ΔGLS). For each node of interest, the primary topology was defined as the species tree recovered by ASTRAL-III, and the alternative topologies were ML trees constrained to recover the most common conflicting bipartitions.

**Identity-by-descent (IBD) analyses.** We performed IBD analyses based on genome-wide SNP data in the genus *Quercus*. By using a same SNP calling and filtering procedure described above (see section "*Orthologous gene identification and nuclear alignment matrix assembly*"). Raw reads of *Quercus* species were trimmed using Trimmomatic v0.39[127], aligned to *Q. robur* reference genome assembly[96] using BWA v0.7.17[128], and called genotypes using GATK v4.2[129]. We applied a strict filtering process to remove all low quality SNPs. We removed all sites located in repetitive regions of the *Q. robur* reference genome[96], and discarded all indels and multiallelic SNPs. We further set genotypes supported by less than four reads as missing data, and deleted SNPs with mean depth <5 or >100, or genotyped in less than half of individuals, or proportion of called heterozygous genotypes >50%. Finally, we obtained 34,250,467 high-quality SNPs for IBD analyses.

We used Beagle v4.1[166] to phase and impute the SNP data, and uncover shared IBD blocks between species. The following parameters were used for IBD analyses in Beagle: window = 100,000; overlap = 10,000; ibdtrim = 100; ibdlod = 5. To compare the recombination rate between IBD blocks and genomic background, we used a genetic map of *Q. robur* developed by Plomion et al.[96]. We smoothed the recombination rate across the genome to 200 kb, and then mapped IBD blocks to the genetic map. For each IBD block, we obtained the recombination rate on middle points of the block, and then used this value as the recombination rate for the whole IBD block.

To test whether the IBDs shared between sections are under selection, we calculated the probability of a selectively neutral haplotype with a given length shared by two sections after introgression. If the IBD blocks were significantly longer than the neutral haplotype, they were most likely maintained by selection after introgression. Following Huerta-Sanchez et al.[91], the probability for each shared IBD block was estimated as: 1- GammaCDF (*L*, shape = 2, rate = lambda), where the GAmmaCDF is the Gamma distribution function and arguments are given in parentheses. The rate parameter lambda was estimated as: lambda = *r* * (*T*/*G*), where *r* is recombination rate, *T* is the time that gene flow occurred, and *G* is the generation time. To calculate the time that gene flow introduced shared IBDs between oak sections, we calculated the genetic divergence (d$_{XY}$) between sections (i.e., *Q. pontica* vs. European and Asian white oaks, North American white oaks vs. section *Virentes* and *Q. sadleriana*) on shared IBD blocks. The estimated mean values of d$_{XY}$ was 0.011–0.018, which was transformed to 2.8–4.5 millions years based on a mutation rate of $2 \times 10^{-9}$ per site per year[11]. Thus, we roughly used 3 million years for the time of introgression. Using a recombination rate of $1 \times 10^{-8}$ estimated from *Q. robur* genetic map (total length of genetic map is 740 cM, and the genome size is 804 Mb)[96], and assuming a generation time of 50 years, we get lambda = $1 \times 10^{-8} \times (3 \times 10^{6}/50) = 6 \times 10^{-4}$. We calculated the probability for each IBD block and corrected multiple testing using Benjamini–Hochberg FDR[161].

To examine whether functional classes of genes were overrepresented in IBD blocks under selection, we performed GO analyses using the R package topGO v2.43.0 (http://www.bioconductor.org/). We applied Fisher's exact test to estimate

the statistical significance of enrichment, and corrected multiple testing by Benjamini–Hochberg FDR[161]. A cutoff of FDR < 0.01 was used to determine the significance of GO enrichment.

**Reporting summary**. Further information on research design is available in the Nature Research Reporting Summary linked to this article.

## Data availability

Short reads of whole-genome sequencing data generated in this study have been deposited in Genbank under accession code PRJNA773751. Alignments of nuclear genes and plastomes generated in this study have been deposited in the Dryad digital data repository (https://doi.org/10.5061/dryad.vq83bk3tc). Previously published genome assemblies are available in Genbank under accession numbers PRJEB14544, PRJEB24056, PRJNA527178, PRJNA433227, PRJEB19898, MG386401, and NC036929. Source data are provided with this paper.

## Code availability

Codes used in this study have been deposited in the Dryad digital data repository (https://doi.org/10.5061/dryad.vq83bk3tc).

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

## Acknowledgements

We are grateful to colleagues who contributed to the research: Al Keuter for assistance with the layout design and select photos of cupules used in Fig. 1; Dylan Burge, Jim Costa, Jie Gao, Yong-Jie Guo, David Hillis, Alejandra Jaramillo, Jason Love, and Ming Qing for providing leaf materials; and Thomas Denk for advice on the fossils. This work was supported by the National Natural Science Foundation of China (grant no. NSFC 31971673 and 32161123003 to B.W.), Guangdong Natural Science Funds for Distinguished Young Scholar (grant no. 2018B030306040 to B.W.), and US National Science Foundation (grant no. 1146102 to P.S.M.).

## Author contributions

B.-F.Z., S.Y., M.K., P.S.M., and B.W. designed the project. B.-F.Z., S.Y., Y.-Y.L., Y.S., X.-Y.C., and Q.-Q.A. collected data. B.-F.Z., S.Y., A.A.C., Y.-Y.L., P.S.M., and B.W. analyzed data. B.-F.Z., S.Y., A.A.C., P.S.M., and B.W. wrote the paper. All authors read and approved the paper.

## Competing interests

The authors declare no competing interests.
