## [Peer Review File · Nature Communications]

Phylogenomic analyses highlight innovation and introgression in the continental radiations of Fagaceae across the Northern HemisphereReviewers' Comments:

Reviewer #1:

Remarks to the Author:

This exceptionally well-written manuscript deals with a number of fundamental evolutionary questions in a novel way. Specifically, Biao-Feng Zhou, Shuai Yuan, Andrew Crowl and co-workers demonstrate using an impressive amount of data how unrelated chloroplast and nuclear genomes both contribute to a holistic understanding of the evolutionary history of a major north temperate group of woody angiosperms, the Fagaceae family. Using time calibrated phylogenies for both genomes and the conflicting signal in these genomes, the authors show that chloroplast genomes encapsulate the shared biogeographic histories of (partly) distantly related groups, whereas the nuclear genome reflects species and higher taxa relationships.

They further show that clade ages from dated phylogenies and diversification rates correlate with major biological innovations possibly triggered by global tectonic-climatic changes. For example, the shift to wind-pollination in oaks, whereas chestnuts and other castaneoids are mainly insect-pollinated, or the shift to hypogeous germination in the HS clade that is often correlated with larger seeds, facilitated major radiations of members of the HS clade under globally cooling conditions. Methodology and analytical approaches are all cutting-edge and the figures accompanying the manuscript are very informative. Figure 2 will most likely be reproduced in text books as it impressively shows how different the chloroplast and nuclear genomes function.

I have only very few comments:

Line 71: Although it is impossible to cite all literature relevant to the evolutionary history of Fagaceae main groups, I think two papers should be cited here:

Sadowski, E.-M., J. U. Hammel, and T. Denk. 2018. Synchrotron X- ray imaging of a dichasium cupule of *Castanopsis* from Eocene Baltic amber. *American Journal of Botany* 105: 2025–2036.

Naryshkina, N. N., and T. A. Evstigneeva. 2020. Fagaceae in the Eocene palynoflora of the South of Primorskii Region: New data on taxonomy and morphology. *Paleontological Journal* 54: 429–439.

Both papers report Eocene findings of *Cyclobalanopsis* from the Northern Hemisphere and partly contradict a hypothesis put forward in another paper by Wilf et al. 2019 (cited in the manuscript).

Line 117: I would say fruits instead of fruit.

Line 329: How would you then explain the presence of sect. *Quercus* pollen in Eocene strata of Hainan, S China? I think the NW origin may be ok, but I was wondering if you have any ideas.

Line 335: Northern Hemisphere instead of northern hemisphere.

Reviewer #2:

Remarks to the Author:

This study by Zhou et al. is a multifaceted examination of the evolutionary history of Fagaceae aimed at understanding the extent and consequences of hybridization during the early diversification of the family and the oak genus (*Quercus*). Overall, I think this is a high-quality study and a strong candidate for publication in *Nature Communications*. The dataset is impressive, the analyses conducted were extensive and rigorous, and the presentation generally does a nice job of synthesizing the results in an interesting and compelling fashion. The economic and ecological importance of the family and particularly the oak genus should also make the study of interest to a broad audience.

Below I outline my major and minor suggestions for revision.

MAJOR COMMENTS

1. At times I think the manuscript is over-written. And by this, I mean (a) the importance of various attributes is exaggerated or based on speculation and (b) attempts at making grandiose statements often results in convoluted, vague, or sometimes not entirely coherent sentences. Below are some examples and comments. I strongly encourage the authors to revise such passages so that overall the presentation is more modest, concrete, and readable.

"Innovations related to seed and pollen dispersal are implicated in triggering waves of continental radiations, while fungal symbioses fortified a competitive edge underground"
This is highly speculative especially since the paper does not include any analyses of trait-dependent diversification.

"This resulted in adaptive introgression, further amplifying global proliferation"
Some analyses were done that suggest positive selection/maintenance of introgression haplotypes, but the leap to "further amplifying global proliferation" seems to much

"Northern Hemisphere forests and shrublands are now dominated by species comprising temperate and subtropical lineages, marking one of the greatest floristic transitions in the vegetation history of the Cenozoic"
NH forests also included many representatives of 'temperate/subtropical' lineages in the early Cenozoic. Undoubtedly there was a major floristic shift after the Eocene, but this sentence reads as vague and empty to me.

"With a minimum age of ca. 80 million years ago (Ma) and precise aging of new fossilized pollen and macrofossils assigned to some modern groups by 50 Ma, the evolution of major lineages appears to be unusually rapid for forest tree species"
One, this doesn't seem that rapid given the ages provided, and two, there are plenty of examples of rapidly radiated woody groups.

The sentence at lines 83-86 seems overly convoluted, as does the sentence from lines 89 to 94.

102: "...the first complete family-wide phylogenetic context"
What does 'complete' mean? This study includes a small fraction of the species diversity. Is it the first nuclear phylogenomic study to include all genera? All genera and all sections of oaks? This seems like another example of trying to stretch the significance when it would be better to just be straightforward and clear.

191 onwards. "Shift to wind-pollination alone did not increase the diversification rate of oak species immediately, but instead served as a predisposed neutral change that later facilitated rapid radiation of this genus during the expansion of seasonal climates (Fig. 1)."
This is speculative. Fine to suggest this as a hypothesis but it is unreasonable to declare it so without any sort of direct tests or evidence. I also don't think the next sentence follows very clearly. I think this could be reframed as a hypothesis with compelling contextual or anecdotal evidence.

231-232: "However, gene flow between modern genera is without precedent"
As you noted previously, phylogenetic studies based on the plastome have suggested gene flow between *Quercus* and related genera (given the geographic structure of the plastid tree). So, I don't think this statement is accurate. But a comprehensive investigation of this using both the nuclear and plastid genomes is novel.

2. I have a few concerns about the dating analyses. One, dating analyses based on a large number of genes can introduce a lot of heterogeneity into analyses that is hard to accommodate. It is now becoming more common to filter your genes for those that (a) match the species-tree topology and (b) are more 'clock-like' (less root-to-tip variation). Since you have such a large number of loci, this time of approach could be beneficial. I understand that many downstream analyses are based on the

dated tree, and so asking to have the dating analyses re-done is a big ask. But this alternative could be tried and compared with the original results to see if there are major differences. Concerning the dated chloroplast tree, you might clarify that the 'ML tree' used as the reference was the CP ML tree (not nuclear). Upon first reading I thought you meant you constrained the plastid analysis to the nuclear topology when dating (which would be problematic) but looking at the Suppl I realized this was not the case. So I would just clarify. I also wonder if it would be worthwhile to do a nuclear dating analysis with only the two calibrations used for the CP analysis. Using different calibration schemes for these datasets seems like it would result in a biased comparison between the two.

MINOR COMMENTS

57: "often repeated seasonal biomes" what does often repeated mean?

69 onward: "Fossil analogs" is perhaps not the best way to put this since 'analog' often refers to independently evolved structures. I would just say "Fossils of Fagaceae are well represented in the Northern Hemisphere..."

106, 553 (and perhaps elsewhere): "coalescent analyses using ASTRAL-III and SVDquartets" Both ASTRAL and SVDquartets are summary methods—they do not model the coalescent, or in other words, they are not based on the coalescent—and so calling them "coalescent-based" approaches is inaccurate

107 change to "for all but a few branches" ?

110: "early-diverging lineages" There are many variations on this throughout the manuscript ('early diverging', 'early branching'). Two branches from a node are equal in age so it doesn't make sense to call one lineage "earlier" with respect to the other(s). This is basically the same issue as calling something "basal". I would encourage the authors to revise this language (change to things like "successively sister" etc). This sentence (109 onward) is also generally convoluted and would benefit from revision.

117 should be "'a' single rounded fruit"?

119: "studies based on sequences derived from RAD-seq datasets and nuclear loci" – this syntax doesn't really make sense

120-122: "Despite phylogenetic congruence across methods, high levels of gene-tree conflict within the nuclear genome were observed, likely due to incomplete lineage sorting (ILS; Supplementary Figs. 2, 3 and 4)." This seems a bit hand-wavy given that the goal of the paper is to integrate deep conflicts to understand ancient hybridization

126: early diverging used again here—revise

134-135: "These events closely follow the Cretaceous-Paleogene (K-Pg) boundary dated at 66

Ma" –15 million years seems like a broad window to say this "closely followed" the KPg

Lines around 156: Do you think any seed traits in this clade might have also conferred greater survivorship following the ecological devastation of the KPg?

175-178: A little confusing whether fagaceae or fungi is being referred to with respect to increased speciation

184: 'coincident' might be a better word here?

291 to 294: Does this also require information on population sizes and, if so, how did you determine those?

323: I don't know if it makes sense for the genomes to be "contributing unique inferences" since the researcher is doing the inferring. "...each contributing unique insights on the complex combination of divergent..." instead?

333: "this level of scrutiny" to "our detection" ?

335-336: This paragraph ender seems a little overblown and vague.

419: remove 's' after 'SCG': 'SGC regions'

Reviewer #3:

Remarks to the Author:

This is an important and exceptionally well-written paper which describes the evolutionary history of the oak family Fagaceae (comprising oak, chestnut, stone oak and beech species) that dominates temperate forests in the northern hemisphere. The impressive, detailed analysis of plastomes and nuclear genomes of representative species across the family builds on previous results obtained for *Quercus* by Manos and others. The robust dated nuclear phylogeny generated tells us when lineages (genera, sub-genera, species) originated and how this ties into ecological innovations (hypogean from epigeal seed germination; wind from insect pollination) and associations (with seed dispersers and mycorrhiza). Most importantly, a comparison of plastome with nuclear phylogenies plus in-depth analysis of gene flow between species and higher lineages provides information on chloroplast capture, historical phylogeography, and adaptive introgression in the family. In conclusion, the authors emphasise the importance of historical introgression to the success of the family.

It is pointed out in the text that many closely related oak species occur in sympatry and hybridise. Consequently, I did wonder how confident we might be that all samples analysed represented the species they were said to be. For example, *Q. robur* and *Q. petraea* are two such species that hybridise to the point that it is very difficult to find plants in the wild that are not hybrids. Are the authors confident that the samples used for these species, and of other species that similarly hybridise extensively, were in fact non-hybrid. If not, how might this affect their findings?

Minor points

lines 497-504. Peculiar. Is there a possible biological reason for the occurrence of these divergent *Q. ilex* plastomes or are they likely artifacts.

676. 'One individual gene trees' ?

716. significantly

722. Suggest modify to 'time that gene flow'

723. Suggest modify to 'time that gene flow'

732. Change to 'block'

Response to Reviewer #1

This exceptionally well-written manuscript deals with a number of fundamental evolutionary questions in a novel way. Specifically, Biao-Feng Zhou, Shuai Yuan, Andrew Crowl and co-workers demonstrate using an impressive amount of data how unrelated chloroplast and nuclear genomes both contribute to a holistic understanding of the evolutionary history of a major north temperate group of woody angiosperms, the Fagaceae family. Using time calibrated phylogenies for both genomes and the conflicting signal in these genomes, the authors show that chloroplast genomes encapsulate the shared biogeographic histories of (partly) distantly related groups, whereas the nuclear genome reflects species and higher taxa relationships.

They further show that clade ages from dated phylogenies and diversification rates correlate with major biological innovations possibly triggered by global tectonic-climatic changes. For example, the shift to wind-pollination in oaks, whereas chestnuts and other castaneoids are mainly insect-pollinated, or the shift to hypogeous germination in the HS clade that is often correlated with larger seeds, facilitated major radiations of members of the HS clade under globally cooling conditions.

Methodology and analytical approaches are all cutting-edge and the figures accompanying the manuscript are very informative. Figure 2 will most likely be reproduced in text books as it impressively shows how different the chloroplast and nuclear genomes function.

I have only very few comments:

Line 71: Although it is impossible to cite all literature relevant to the evolutionary history of Fagaceae main groups, I think two papers should be cited here:

*Sadowski, E.-M., J. U. Hammel, and T. Denk. 2018. Synchrotron X- ray imaging of a dichasium cupule of *Castanopsis* from Eocene Baltic amber. *American Journal of Botany* 105: 2025–2036.*

*Naryshkina, N. N., and T. A. Evstigneeva. 2020. Fagaceae in the Eocene palynoflora of the South of Primorskii Region: New data on taxonomy and morphology. *Paleontological Journal* 54: 429–439.*

*Both papers report Eocene findings of *Cyclobalanopsis* from the Northern Hemisphere and partly contradict a hypothesis put forward in another paper by Wilf et al. 2019 (cited in the manuscript).*

----Thank you. We have added these two literatures, although addressing the point on the Wilf et al. (2019) paper would require a separate review on the biogeographic history of Fagaceae. Lines 70, 874-879.

Line 117: I would say fruits instead of fruit.

----We have changed this to read “a single rounded fruit”. Line 117.

*Line 329: How would you then explain the presence of sect. *Quercus* pollen in Eocene strata of Hainan, S China? I think the NW origin may be ok, but I was wondering if you have any ideas.*

*----Interesting challenge. We made some minor changes to the text to reflect the potential for an early presence of section *Quercus* at tropical latitudes in Asia based on*

studies of Hofmann et al data (2019) from Eocene strata in Hainan. Our divergence time estimates based on nuclear data for NW/OW white oak split is roughly as early as 26 Ma (mid-Oligocene), so a late Eocene presence of section *Quercus* is reasonable, except for the surprisingly tropical extension of its distribution in Hainan. Lines 325, 366-368.

The earlier presence of section could be explained by broadly distributed ancestral section *Quercus* or Pacific lineage that spanned the OW/NW through Beringia at a time when northwestern North America and northeastern Asia were more or less continuous areas with a shared flora. A similar pattern of expansion and contraction has been invoked to explain the fossil history of *Fagus*, which is now extinct from northwestern North America, but extant in east Asia (Denk & Grimm 2009). The paleo distribution of Fagaceae in and around the high latitudes associated with the Beringian Land Bridge and into more southward amphi-Pacific areas is clearly at odds with the modern distribution of taxa.

As Hainan became progressively more tropical (extant Fagaceae include *Lithocarpus*, *Castanopsis*, *Trigonobalanus*, sections *Cyclobalanopsis*, *Cerris*, and *Ilex*), subtropical and temperate Fagaceae like section *Quercus* may have gone extinct (Zhu 2016). A second expansion of section *Quercus* derived from a NW temperate ancestor shared with sect *Quercus* subsect *Albae* could explain a more recent dispersion of Eurasian white oaks across Eurasia during the expansion of seasonal climates in the Oligocene.

Hofmann CC, Kodrul TM, Liu XY, Jin JH. 2019. Scanning electron microscopy investigations of middle to late Eocene pollen from the Changchang Basin (Hainan island, south China) – Insights into the paleobiogeography and fossil history of *Juglans*, *Fagus*, *Lagerstroemia*, *Mortoniiodendron*, *Cornus*, *Nyssa*, *Symplocos* and some *Icacinaceae* in SE Asia. *Review of Palaeobotany and Palynology* 265: 41–61.

Denk T, Grimm GW. 2009. The biogeographic history of beech trees. *Review of Palaeobotany and Palynology* 158: 83–100.

Zhu H. 2016. Biogeographical evidences help revealing the origin of Hainan Island. *PLoS ONE* 11: e0151941.

Line 335: Northern Hemisphere instead of northern hemisphere.

----This change has been made. Line 334.

Reviewer #2 (Remarks to the Author):

This study by Zhou et al. is a multifaceted examination of the evolutionary history of Fagaceae aimed at understanding the extent and consequences of hybridization during the early diversification of the family and the oak genus (Quercus). Overall, I think this is a high-quality study and a strong candidate for publication in Nature Communications. The dataset is impressive, the analyses conducted were extensive and rigorous, and the presentation generally does a nice job of synthesizing the results in an interesting and compelling fashion. The economic and ecological importance of the family and particularly the oak genus should also make the study of interest to a broad audience.

Below I outline my major and minor suggestions for revision.

MAJOR COMMENTS

1. At times I think the manuscript is over-written. And by this, I mean (a) the importance of various attributes is exaggerated or based on speculation and (b) attempts at making grandiose statements often results in convoluted, vague, or sometimes not entirely coherent sentences. Below are some examples and comments. I strongly encourage the authors to revise such passages so that overall the presentation is more modest, concrete, and readable.

*“Innovations related to seed and pollen dispersal are implicated in triggering waves of continental radiations, while fungal symbioses fortified a competitive edge underground”
This is highly speculative especially since the paper does not include any analyses of trait-dependent diversification.*

---We have edited this statement to focus only on seed dispersal, which was found to be associated with a spike in diversification in our study. We removed mention of pollen dispersal and fungal symbioses. Lines 41-44.

*“This resulted in adaptive introgression, further amplifying global proliferation”
Some analyses were done that suggest positive selection/maintenance of introgression haplotypes, but the leap to “further amplifying global proliferation” seems to much.*

---We have removed this problematic phrase and have rewritten this sentence to focus specifically on the Eurasian white oak clade. Diversification analyses identified this clade as having an increase in net diversification. We feel the link between positive selection and increased diversification in this clade is sufficient to make such a statement. Lines 48-49.

“Northern Hemisphere forests and shrublands are now dominated by species comprising temperate and subtropical lineages, marking one of the greatest floristic transitions in the vegetation history of the Cenozoic”

NH forests also included many representatives of ‘temperate/subtropical’ lineages in the early Cenozoic. Undoubtedly there was a major floristic shift after the Eocene, but this sentence reads as vague and empty to me.

---We have toned this statement down to read “one of the major floristic transitions”. Line 53.

“With a minimum age of ca. 80 million years ago (Ma) and precise aging of new fossilized pollen and macrofossils assigned to some modern groups by 50 Ma, the evolution of major lineages appears to be unusually rapid for forest tree species”

One, this doesn’t seem that rapid given the ages provided, and two, there are plenty of examples of rapidly radiated woody groups.

---We appreciate this comment and have revised the text to instead focus on the timing of the diversification of the HS clade of Fagaceae, which we found to be associated with an increase in net diversification rate (Lines 72-76). Fossil taxa ascribed to those genera fall within a much narrower window of time. We are using the rich fossil record of

Fagaceae to suggest that macroevolutionary change (pollen, cupule/fruit) was rapid in deep time, in contrast to the general findings that woody lineages evolve more slowly than herbaceous lineages using relative rate analysis across sister clades. Life history correlates such as generation time also have figured into this broader question (see Smith & Beaulieu 2009)

This is an interesting topic that intersects with ideas on the role of life history, climate niche evolution, and correlates with net diversification rates. Following the publication of Smith and Donoghue (2008, cited here) and Smith & Beaulieu (2009), some literature suggests that secondary woodiness is a prerequisite to higher speciation rates in isolated niche space, such as islands and continental sky islands. While there is some evidence for rapid diversification in woody clades such as temperate *Quercus* (Hipp et al. 2020, cited here and detected in this study as well) and tropical *Inga* (Fabaceae, Richardson et al. 2001), slower rates of evolution as a function life history, limited sequence divergence or net diversification rates remain associated with tree lineages. More recent analyses suggest that plant size, another proxy for arborescence, also is correlated with lower diversification rates across the angiosperms (Boucher et al. 2017).

Boucher FC, Verboom GA, Musker S, Ellis AG (2017) Plant size: a key determinant of diversification? *New Phytologist* 216: 24-31.

Nürk NM, Atchison GW, Hughes CE (2019) Island woodiness underpins accelerated disparification in plant radiations. *New Phytologist* 224: 518-531.

Richardson JE, Pennington RT, Pennington TD, Hollingsworth PM (2001) Rapid diversification of a species-rich genus of Neotropical rain forest trees. *Science* 293: 2242-2245

Smith SA and Beaulieu JM (2009) Life-history influences rates of climatic niche evolution in flowering plants. *Proceedings of the Royal Society B* 276: 4345–4352

The sentence at lines 83-86 seems overly convoluted, as does the sentence from lines 89 to 94.

----We have attempted to clarify and reduce the length of these sentences. Lines 84-86, 92.

102: “...the first complete family-wide phylogenetic context”

What does ‘complete’ mean? This study includes a small fraction of the species diversity. Is it the first nuclear phylogenomic study to include all genera? All genera and all sections of oaks? This seems like another example of trying to stretch the significance when it would be better to just be straightforward and clear.

----We have changed this to simply read “within a broad phylogenetic context.” Line 102.

191 onwards. “Shift to wind-pollination alone did not increase the diversification rate of oak species immediately, but instead served as a predisposed neutral change that later

facilitated rapid radiation of this genus during the expansion of seasonal climates (Fig. 1)."

This is speculative. Fine to suggest this as a hypothesis but it is unreasonable to declare it so without any sort of direct tests or evidence. I also don't think the next sentence follows very clearly. I think this could be reframed as a hypothesis with compelling contextual or anecdotal evidence.

----We agree and have rephrased this as a hypothesis. Lines 190, 193-194.

231-232: *"However, gene flow between modern genera is without precedent"*

*As you noted previously, phylogenetic studies based on the plastome have suggested gene flow between *Quercus* and related genera (given the geographic structure of the plastid tree). So, I don't think this statement is accurate. But a comprehensive investigation of this using both the nuclear and plastid genomes is novel.*

----We have clarified this to read, "inference of gene flow between modern genera has been based solely on plastome data." Lines 230-231.

2. *I have a few concerns about the dating analyses. One, dating analyses based on a large number of genes can introduce a lot of heterogeneity into analyses that is hard to accommodate. It is now becoming more common to filter your genes for those that (a) match the species-tree topology and (b) are more 'clock-like' (less root-to-tip variation). Since you have such a large number of loci, this time of approach could be beneficial. I understand that many downstream analyses are based on the dated tree, and so asking to have the dating analyses re-done is a big ask. But this alternative could be tried and compared with the original results to see if there are major differences.*

--- Thank you for raising this important point. We have applied two "gene-shopping" methods to identify genes with the best information for dating. First, we used SortaData (Smith *et al.* 2018) to filter 212 (top 90th percentile) most clock-like loci by considering clock-like as the primary criterion, followed by tree-like and tree length. Second, we calculated the Robinson-Foulds (RF) distance between gene trees and the reference tree following Johns *et al.* (2018), and retained 212 loci with the least RF distance and greater concordant phylogenetic signals. By using three different topologies as reference trees (see above), we generated six reduced datasets. The divergence time estimated on the 2124 genes were almost identical to the six reduced data sets (Pearson's correlation coefficient = 0.991-0.995, $P < 2e^{-16}$; Supplementary Fig. 15). These new approaches have been described in the revised Methods (Lines 566-576), a new Supplementary Table 11 and a new Supplementary Fig. S15.

Concerning the dated chloroplast tree, you might clarify that the 'ML tree' used as the reference was the CP ML tree (not nuclear). Upon first reading I thought you meant you constrained the plastid analysis to the nuclear topology when dating (which would be problematic) but looking at the Suppl I realized this was not the case. So I would just clarify.

---- We have clarified this to "plastome ML tree". Line 578.

I also wonder if it would be worthwhile to do a nuclear dating analysis with only the two calibrations used for the CP analysis. Using different calibration schemes for these datasets seems like it would result in a biased comparison between the two.

---We have dated the nuclear tree by using the same two calibrations for plastome tree (Lines 585-587). This result was presented in updated Supplementary Fig. 5g-i.

MINOR COMMENTS

57: “often repeated seasonal biomes” what does often repeated mean?

---This was intended to plant the seed that similar biomes present the opportunity for ecological convergence in Fagaceae. The text has been simplified. Lines 56-57.

69 onward: “Fossil analogs” is perhaps not the best way to put this since ‘analog’ often refers to independently evolved structures. I would just say “Fossils of Fagaceae are well represented in the Northern Hemisphere...”

---Good point. This change has been made. Line 69.

106, 553 (and perhaps elsewhere): “coalescent analyses using ASTRAL-III and SVDquartets” Both ASTRAL and SVDquartets are summary methods—they do not model the coalescent, or in other words, they are not based on the coalescent—and so calling them “coalescent-based” approaches is inaccurate

---We have remedied this by either removing the term “coalescent analyses” or changing to read “species-tree analyses” throughout the manuscript. Lines 106, 530 and 563.

107 change to “for all but a few branches” ?

---This sentence has been revised. Lines 107.

110: “early-diverging lineages” There are many variations on this throughout the manuscript (‘early diverging’, ‘early branching’). Two branches from a node are equal in age so it doesn’t make sense to call one lineage “earlier” with respect to the other(s). This is basically the same issue as calling something “basal”. I would encourage the authors to revise this language (change to things like “successively sister” etc). This sentence (109 onward) is also generally convoluted and would benefit from revision.

---All mention of “early-diverging lineages” has been removed or replaced with “successively sister lineages”. Lines 110, 126, 149, 160 and 204.

117 should be “‘a’ single rounded fruit”?

---This change has been made. Lines 117.

119: “studies based on sequences derived from RAD-seq datasets and nuclear loci” – this syntax doesn’t really make sense

---This sentence has been amended to read, “...previous studies based on RAD-seq data.” Lines 119.

120-122: “Despite phylogenetic congruence across methods, high levels of gene-tree conflict within the nuclear genome were observed, likely due to incomplete lineage

sorting (ILS; Supplementary Figs. 2, 3 and 4).” This seems a bit hand-wavy given that the goal of the paper is to integrate deep conflicts to understand ancient hybridization

---This observation refers to the relationships among *Quercus*, *Notholithocarpus*, *Lithocarpus*, and *Chrysolepis* which we found to be in conflict across nuclear gene trees. Not surprisingly, ILS and hybridization present challenges for estimating the Fagaceae tree of life. For this particular set of relationships and narrow time frame, a stochastic pattern of gene tree resolution is consistent with ILS. These results were detailed in Supplementary Figs. 2, 3 and 4, and briefly referred in main text (lines 119-122)

126: *early diverging used again here—revise*

---This sentence has been revised. Line 126.

134-135: “These events closely follow the Cretaceous-Paleogene (K-Pg) boundary dated at 66 Ma” –15 million years seems like a broad window to say this “closely followed” the KPg

---‘closely’ has been removed from this sentence to state that these events simply followed the KPg. Line 133.

Lines around 156: *Do you think any seed traits in this clade might have also conferred greater survivorship following the ecological devastation of the KPg?*

---Interesting question, and much appreciated. But difficult to add anything concrete on the subject. All seeds produced by Fagaceae are known to have a short life span in the soil. As there is no seed bank, the seeds produced in one season are the only source of recruitment. Recent ecological work suggests that seed masting is the main driver for seedling recruitment in forest trees, highlighting the strong selection pressure of seed predators and episodic nature of successful recruitment. While the shift to hypogeous cotyledons appears to be associated with diversifying potential dispersers, its evolution has been linked to increased survivorship over short time scales. Regarding other traits, anecdotal evidence suggests that selection in fruit/cupule evolution in *Castanea* + *Castanopsis* was driven to protect seeds through development whereas enhanced dispersal capacity is a general theme in *Lithocarpus* and *Quercus* (see Fig 1 images of spiny cupules vs scaly cupules surrounding or subtending nuts), with secondarily evolved exceptions scattered in both genera to protect the seed, and one notable origin of the acorn fruit type in *Castanopsis*.

175-178: *A little confusing whether fagaceae or fungi is being referred to with respect to increased speciation*

---We have clarified that it is species radiations of Fagaceae. Lines 176-177.

184: *‘coincident’ might be a better word here?*

---This change has been made. Line 183.

291 to 294: Does this also require information on population sizes and, if so, how did you determine those?

---In the method developed by Huerta-Sanchez *et al.* (2014), the probability of maintaining selectively neutral haplotypes of a given length in both oak sections after

introgression was determined by the recombination rate, the time that gene flow occurred and generation time. The effective population size was not included in the function. See details in Methods (Lines 724-733).

323: *I don't know if it makes sense for the genomes to be "contributing unique inferences" since the researcher is doing the inferring. "...each contributing unique insights on the complex combination of divergent..." instead?*

----Agreed. This change has been made. Line 322.

333: *"this level of scrutiny" to "our detection" ?*

----This change has been made. Lines 331-332.

335-336: *This paragraph ender seems a little overblown and vague.*

----We have edited this to simply read, "The ecological implications of these biotic exchanges of keystone lineages await future study." Lines 334-335.

419: *remove 's' after 'SCG': 'SGC regions'*

----This change has been made. Line 417.

Reviewer #3 (Remarks to the Author):

*This is an important and exceptionally well-written paper which describes the evolutionary history of the oak family Fagaceae (comprising oak, chestnut, stone oak and beech species) that dominates temperate forests in the northern hemisphere. The impressive, detailed analysis of plastomes and nuclear genomes of representative species across the family builds on previous results obtained for *Quercus* by Manos and others. The robust dated nuclear phylogeny generated tells us when lineages (genera, sub-genera, species) originated and how this ties into ecological innovations (hypogeal from epigeal seed germination; wind from insect pollination) and associations (with seed dispersers and mycorrhiza). Most importantly, a comparison of plastome with nuclear phylogenies plus in-depth analysis of gene flow between species and higher lineages provides information on chloroplast capture, historical phylogeography, and adaptive introgression in the family. In conclusion, the authors emphasise the importance of historical introgression to the success of the family.*

*It is pointed out in the text that many closely related oak species occur in sympatry and hybridise. Consequently, I did wonder how confident we might be that all samples analysed represented the species they were said to be. For example, *Q. robur* and *Q. petraea* are two such species that hybridise to the point that it is very difficult to find plants in the wild that are not hybrids. Are the authors confident that the samples used for these species, and of other species that similarly hybridise extensively, were in fact non-hybrid. If not, how might this affect their findings?*

---- Good point. The plant material we used in the study was derived from a combination of collections made from natural populations and cultivated plants grown from wild-collected seed. We have updated Supplementary Table 1 to provide data on our voucher specimens.

Where possible, our joint expertise in Fagaceae taxonomy guided the selection of species, in addition to generally including species that are taxonomically well-understood and easily identified. However, it is true that later generation hybrids may express phenotypes that often converge on one parental species. While we do present evidence of more recent hybridization between several morphologically typical species (see Figure 3A), we believe the impact on our findings at deeper phylogenetic levels is minimal. Our tests basically support the mosaic nature of closely related oak genomes, e.g., sympatric white oaks, initially suggested by analyses using RAD-seq data (Hipp et al. 2020, cited here). And despite gene flow involving the tips of the tree, we detected a high degree of coalescence for the gene tree histories that support the common ancestry of the genera and sections of *Quercus*. From our analyses, it seems that ILS (Incomplete Lineage Sorting) is the main challenge for estimating the species tree in the crown clade of Fagaceae.

Minor points

lines 497-504. *Peculiar. Is there a possible biological reason for the occurrence of these divergent Q. ilex plastomes or are they likely artifacts.*

---- One potential biological explanation for the two highly divergent *Q. ilex* plastomes could be ancient haplotypes retained in this species. However, previous analyses with extensive sampling spanning the geographic distribution of *Q. ilex* placed this species within a clade formed by Eurasian oaks and genera *Castanea* and *Castanopsis* based on plastid data (Simeone et al. 2016). Therefore, we concluded that the two *Q. ilex* plastomes in question are most likely artifacts, although more sampling is required to increase our confidence of that assertion.

676. *'One individual gene trees' ?*

----This has been changed to read "One individual gene tree". Line 686.

716. *significantly*

----This change has been made. Line 726.

722. *Suggest modify to 'time that gene flow'*

----This change has been made. Line 732.

723. *Suggest modify to 'time that gene flow'*

----This change has been made. Line 733.

732. *Change to 'block'*

----This change has been made. Line 742.

Reviewers' Comments:

Reviewer #1:

Remarks to the Author:

The manuscript has further improved after addressing the reviewers comments and I recommend accepting it for publication.

Reviewer #2:

Remarks to the Author:

This revision by Zhou et al. does an excellent job of revising and responding to the reviewer comments (including my own). I am very satisfied with how they responding to my comments regarding different aspects of the text, and also appreciate the thorough additional dating analyses conducted in responses to my suggestion. I have one comment/critique related to the revised sentence starting on Line 72 (described below), that the authors might consider for a slight revision. But otherwise I think the manuscript is in excellent shape and now suitable for publication in Nature Communications.

Line 72 onward. I appreciate the reply to my original comment. But I think the sentence still reads a bit awkwardly and I think it still has issues. The information presented here does not itself suggest anything about rapid morphological change. Why not say more specifically that although the family/stem is ca. 80 million years old, divergence of crown groups/the HS clade appears to have occurred rapidly in the early Cenozoic, suggesting rapid morphological evolution?

I understand your position and agree that both evidence and expectations suggest generally slower evolutionary rates in long-lived lineages such as trees. But the 30-million-year window you present in the sentence is not a particularly narrow window of time, even for lineages we would typically expect to be slowly evolving. But it seems that you could revise to more clearly and accurately support your position—the diversification of the specific clade(s) in question occurred in a narrower window.

In this context, you might also cite this recent paper suggesting a general link between phylogenomic conflict (as a signature of rapid diversification) and rapid phenotypic evolution, which might be relevant in your case:

Parins-Fukuchi, Caroline, Gregory W. Stull, and Stephen A. Smith. "Phylogenomic conflict coincides with rapid morphological innovation." *Proceedings of the National Academy of Sciences* 118, no. 19 (2021).

Response to Reviewer #1

Reviewer #1 (Remarks to the Author):

The manuscript has further improved after addressing the reviewers comments and I recommend accepting it for publication.

---Many thanks for your insightful comments on a previous draft.

Response to Reviewer #2

Reviewer #2 (Remarks to the Author):

This revision by Zhou et al. does an excellent job of revising and responding to the reviewer comments (including my own). I am very satisfied with how they responding to my comments regarding different aspects of the text, and also appreciate the thorough additional dating analyses conducted in responses to my suggestion. I have one comment/critique related to the revised sentence starting on Line 72 (described below), that the authors might consider for a slight revision. But otherwise I think the manuscript is in excellent shape and now suitable for publication in Nature Communications.

Line 72 onward. I appreciate the reply to my original comment. But I think the sentence still reads a bit awkwardly and I think it still has issues. The information presented here does not itself suggest anything about rapid morphological change. Why not say more specifically that although the family/stem is ca. 80 million years old, divergence of crown groups/the HS clade appears to have occurred rapidly in the early Cenozoic, suggesting rapid morphological evolution?

I understand you position and agree that both evidence and expectations suggest generally slower evolutionary rates in long-lived lineages such as trees. But the 30-million-year window you present in the sentence is not a particularly narrow window of time, even for lineages we would typically expect to be slowly evolving. But it seems that you could revise to more clearly and accurately support your position—the diversification of the specific clade(s) in question occurred in a narrower window.

In this context, you might also cite this recent paper suggesting a general link between phylogenomic conflict (as a signature of rapid diversification) and rapid phenotypic evolution, which might be relevant in your case:

*Parins-Fukuchi, Caroline, Gregory W. Stull, and Stephen A. Smith. "Phylogenomic conflict coincides with rapid morphological innovation." *Proceedings of the National Academy of Sciences* 118, no. 19 (2021).*

---Thank you for raising this point. We have revised the statement as follows: “While a minimum divergence age of ca. 80 million years ago (Ma) is estimated for the family, divergence of crown groups appears to have occurred rapidly in the early Cenozoic, suggesting the potential for rapid morphological change in forest tree species⁴⁴⁻⁴⁷”. The suggested reference has been cited. Lines 78-81 and 945-947.